# Impact of a Primary Care Antimicrobial Stewardship Program on Bacterial Resistance Control and Ecological Imprint in Urinary Tract Infections

**DOI:** 10.3390/antibiotics11121776

**Published:** 2022-12-08

**Authors:** Alfredo Jover-Sáenz, María Ramírez-Hidalgo, Alba Bellés Bellés, Esther Ribes Murillo, Meritxell Batlle Bosch, José Cayado Cabanillas, Sònia Garrido-Calvo, María Isabel Gracia Vilas, Laura Gros Navés, María Jesús Javierre Caudevilla, Alba Mari López, Lidia Montull Navarro, Mireia Ortiz Valls, Gemma Terrer Manrique, Pilar Vaqué Castilla, José Javier Ichart Tomás, Elena Justribó Sánchez, Ester Andreu Mayor, Joan Carrera Guiu, Roser Martorell Solé, Silvia Pallàs Satué, Mireia Saura Codina, Ana Vena Martínez, José Manuel Albalat Samper, Susana Cano Marrón, Isabel Chacón Domínguez, David de la Rica Escuin, María José Estadella Servalls, Ana M. Figueres Farreny, Sonia Minguet Vidal, Luís Miguel Montaña Esteban, Josep Otal Bareche, Mercè Pallerola Planes, Francesc Pujol Aymerich, Andrés Rodríguez Garrocho, Antoni Solé Curcó, Francisca Toribio Redondo, María Cruz Urgelés Castillón, Juan Valgañon Palacios, Joan Torres-Puig-gros

**Affiliations:** 1Unidad Territorial Infección Nosocomial (UTIN), Hospital Universitari Arnau de Vilanova de Lleida (HUAV), Institut de Recerca Biomèdica (IRBLleida), 25198 Lleida, Spain; 2Sección de Microbiología, HUAV, 25198 Lleida, Spain; 3Unidad de Farmacia de Atención Primaria, Institut Català de la Salut (ICS), 25007 Lleida, Spain; 4Equipo de Atención Priamaria (EAP) Les Borges Blanques, 25400 Lleida, Spain; 5EAP Pla d’Urgell, 25001 Lleida, Spain; 6EAP Balàfia-Pardinyes, 25005 Lleida, Spain; 7EAP Rambla de Ferran, 25007 Lleida, Spain; 8EAP Lleida Rural Nord, 25110 Lleida, Spain; 9Centre Penitenciari de Ponent, 25199 Lleida, Spain; 10EAP Eixample, 25006 Lleida, Spain; 11EAP Primer de Maig, 25002 Lleida, Spain; 12Servicio de Urgencias. HUAV, 25198 Lleida, Spain; 13Centre Urgències Atenció Primària (CUAP), 25004 Lleida, Spain; 14Col·legi Oficial de Podòlegs, 25001 Lleida, Spain; 15Col·legi Oficial de Odontòlegs, 25002 Lleida, Spain; 16EAP Cervera,25200 Lleida, Spain; 17Col·legi Farmacèutics de Lleida, 25007 Lleida, Spain; 18Servei de Geriatria, Hospital Universitari Santa Maria, 25198 Lleida, Spain; 19EAP Ponts, 25740 Lleida, Spain; 20EAP Onze de Setembre, 25005 Lleida, Spain; 21EAP Bellpuig, 25250 Lleida, Spain; 22EAP Artesa de Segre, 25730 Lleida, Spain; 23EAP Cappont, 25001 Lleida, Spain; 24EAP Almacelles, 25100 Lleida, Spain; 25EAP Agramunt, 25310 Lleida, Spain; 26EAP Seròs, 25183 Lleida, Spain; 27EAP Lleida Rural Sud, 25171 Lleida, Spain; 28EAP Balaguer, 25600 Lleida, Spain; 29EAP Alcarràs, 25180 Lleida, Spain; 30EAP Tàrrega, 25300 Lleida, Spain; 31EAP Alfarràs-Almenar, 25120 Lleida, Spain; 32EAP Bordeta—Magraners, 25001 Lleida, Spain; 33EAP La Granadella, 25177 Lleida, Spain; 34Departament de Salut Pública, Universitat de Lleida (UdL), 25006 Lleida, Spain

**Keywords:** antimicrobial stewardship, use antimicrobials, multidrug-resistant microorganisms, community onset, epidemiology

## Abstract

Antimicrobial stewardship programs (ASPs) are a central component in reducing the overprescription of unnecessary antibiotics, with multiple studies showing benefits in the reduction of bacterial resistance. Less commonly, ASPs have been performed in outpatient settings, but there is a lack of available data in these settings. We implemented an ASP in a large regional outpatient setting to assess its feasibility and effectiveness. Over a 5-year post-implementation period, compared to the pre-intervention period, a significant reduction in antibiotic prescription occurred, with a reduction in resistance in *E. coli* urinary isolates. ASP activities also were found to be cost-effective, with a reduction in medication prescription.

## 1. Introduction

Antimicrobial resistance is one of the main global public health threats [1,2,3]. The overuse of antibiotics, particularly broad-spectrum antibiotics, is one of the greatest contributions to the increase in resistance rates, with an inappropriate prescription in approximately 50% of patients [4,5]. Antimicrobial stewardship programs (ASPs) combining infection control measures play a crucial role in reducing community resistance rates. There is sufficient scientific evidence to recommend the universal implementation of these programs [6,7,8,9]. ASPs have proven to be highly effective in hospital facilities, providing better clinical courses and reducing ecological impacts [10,11]. However, in primary care (PC) settings, responsible for 75% of all antibiotic prescriptions [12], and thus having a large impact on microbial resistance, data regarding the implementation of ASPs and long-term results are limited [13,14,15].

In recent years, there has been a continuous increase in the rates of Enterobacterales with extended-spectrum β-lactamase (ESBL) around the world, especially in Asia [16]. In a recent review of their global epidemiology, the prevalence of CTX-M ESBLs was found to have increased over time in all geographic regions, especially in community isolates [17]. In Spain, data extracted from the Study for Monitoring Antimicrobial Resistance Trends (SMART) from 2011 to 2015 and 2016 to 2017 revealed a slight increase in ESBL-producing *Escherichia coli* isolates (7.6% to 8.1%) and a higher increase in *Klebsiella pneumoniae* from 18.6% in 2015 to 25.4% in 2016–2017 [18,19]. Recent meta-analyses have shown a direct relationship between exposure to certain antimicrobial families and the microbiological resistance of these bacteria [20,21], and thus it is a priority to use the narrowest-spectrum agent [22,23]. 

Previous research from our group, carried out in hospital facilities between 2013 and 2017, showed a pronounced downward trend in the incidence densities (ID) of nosocomial multidrug-resistant microorganisms (MDRM) after a period of ASP intervention (RR 0.78, 95% CI 0.73 to 0.84, *p* < 0.001), associated with an overall reduction in antimicrobial consumption of 5.7%, particularly for fluoroquinolones (*p* < 0.015) [24]. We aimed to expand the scope of a successful ASP to the regional outpatient setting with a specific focus on reducing the duration of antibiotics for urinary tract infection (UTI) and reducing prescriptions of fluroquinolones, cephalosporins, and amoxicillin-clavulanic acid. Our hypothesis was that this program would be feasible in a large healthcare network and would result in meaningful decreases in healthcare costs and in antimicrobial resistance in *E. coli* and *K. pneumoniae*. 

## 2. Materials and Methods

### 2.1. Design, Setting, and Study Periods

This pre- and post-intervention quasi-experimental study was carried out in the region of Lleida, belonging to the public health network of Catalonia (CatSalut), Spain, during the period from January 2014 to December 2021 (5 years); we carried out the intervention in January 2017. General practitioners and pediatricians assist a reference population of 340,000 inhabitants, in 23 integrated primary care centers in the region, in direct coordination with a regional microbiology laboratory and a level 3 reference hospital.

In 2016, the region’s infections and antibiotic policy board, constituting professionals of different specialties and categories, and administration staff, launched a specific ASP for the community setting [25], joining a program that already encompassed other areas (acute hospitals, long-term care facilities, and geriatric residences) called P-ILEHRDA. The design was created considering the consensus ASP document published by the Spanish Society of Infectious Diseases and Clinical Microbiology, adapted to the characteristics of the region [26]. It received administrative support for its implementation.

The implementation of the ASP was based on the inter- and multidisciplinary action of referring professional teams in each primary care center, formed by at least one general practitioner, a nurse, and a pediatrician, as well as a coordinating technical team consisting of general practitioners, infectious disease specialists, pediatricians, a microbiologist, a primary care pharmacist, a community pharmacist, a geriatrician, an emergency physician, a podiatrist, and a dentist. The clinical referents were chosen for their interest, knowledge, experience, analytical capacity, relationship with the team, and skills in training. 

The program included the following educational and training actions: (1) creation and periodic updating of regional protocols of antibiotic treatment for the most prevalent infectious diseases (urinary tract, respiratory, skin and soft tissue, and dental infections), based on scientific evidence; (2) development of a mobile application (ProAPP Lleida), free to download, in order to display the mentioned protocols, which was also available on the intranet of the institution; (3) general and specific updated training of professionals through courses, sessions, workshops, or seminars carried out a minimum of 3 times a year; (4) daily review of all positive microbiological results and weekly review of antibiotic prescriptions of each center, except weekends and holidays; (5) daily written non-imposed advice for professionals on computerized SAP “Systems, Applications, Products in Data Processing” or E-cap “Estació Clínica d’Atenció Primària” medical history, advice on site or by telephone—the consultations emphasized the suitability of empirical therapy, targeted treatments, dose adjustments, de-escalating, and shortening duration, toxicity, or interactions; (6) annual consumption monitoring reports, ID of MDRM, and local microbiological sensitivity. The actions were not subject to extraordinary remuneration to implicated professionals. The workflow diagram is shown in Figure 1.

No restrictive measures were imposed on prescriptions in any of the periods studied. The type of recommendation was prospectively compiled to study the incidence over time. Consultancies were interrupted in 2020, due to the SARS-CoV-2 pandemic.

### 2.2. Sources of Information

Antibiotic prescription and microbiological resistance information was obtained from regional dispensing data and a computerized data management system, respectively. For the temporal statistical analysis, the number of patients with medical cards per semester was collected.

### 2.3. Measurement of Consumption and Microbiological and Economic Impact

The primary outcome was the change in global antimicrobial community consumption, especially the non-recommended antibiotics (NRA), due to their associated resistance (quinolones, cephalosporins, and co-amoxiclav), by semester, in the intervention period 2017–2021, compared to a previous reference period. The secondary outcome was the trend in the evolution of common *Enterobacteriaceae* (*E. coli* and *K. pneumoniae*) in their resistance to ciprofloxacin or co-amoxiclav, or ESBL production. The third outcome was the reduction in expenses attributable to the results of the ASP in antibiotic consumption.

### 2.4. Evaluation Methods

To evaluate the consumption of antimicrobials, the Anatomical Therapeutic Chemical Classification and Defined Daily Dose System (ATC/DDD) instituted by the World Health Organization (WHO) (https://www.who.int/tools/atc-ddd-toolkit/about-ddd) (accessed on 6 December 2022) was used and expressed as the number of DDDs per 1000 inhabitants with medical cards per day (DID). DDDs correspond to the assumed average maintenance dose per day for a drug used for its main indication in adults.

The evolutionary impact of resistance was assessed by calculating the ID per 1000 inhabitants per day of the mentioned bacteria, semi-annually, which was similar to antibiotic consumption estimations. To perform the calculation, only one positive culture per person and semester was counted. We assume a delay of 6 months between intervention, implementation, and any associated changes in resistance, as noted in some articles [27]. For this reason, the antimicrobial resistance analysis was extended by up to 6 months from the end of the study period. The percentage of resistance was identified as the resistant samples among the total number of antibiograms performed. For the definition of MDRM, the international standard criteria proposed in consensus by Magiorakos et al. [28] were used. The identification of new cases, from a single clinical sample, was carried out by the Microbiology Department, which determined antibiotic resistance according to the International Laboratory Standards (ISL) [29] and European Committee on Antimicrobial Susceptibility Testing (EUCAST) recommendations [30].

The Pharmacy Service of the Catalan Health Service (Catsalut) evaluated the consumption data (expressed in euros, EUR) obtained from the specific electronic prescription software used in the health system, according to the standard fee in the study periods.

### 2.5. Statistical Analysis 

Continuous quantitative variables were expressed as mean ± standard deviation (SD) and categorical variables as frequencies and percentages (%). A graphical representation of antibiotic consumption and resistance over time was obtained through pre- and post-intervention histograms highlighting the cut-off point. Comparison between IDs was expressed as the relative risk (RR), and the odds ratio (OR) was used to evaluate resistance rates. To measure the impact of ID, the attributable risk (absolute effect) and the preventable fraction (relative effect) expressed in percentage were used. The analysis of the effects attributable to the intervention was calculated by comparing the pre- and post-intervention periods and also at three cut-off points at the beginning, middle, and end of the intervention period. The temporal trends in the pre- and post-intervention periods were analyzed through the linear trend chi-square test. Changes in quantitative variables such as ID were analyzed through the Student–Fisher t test and single-factor ANOVA. Data were analyzed using EPIDAT (version 3.1) of the Pan American Health Organization and are presented with 95% confidence intervals (95% CI). Significance was defined as *p* < 0.05.

## 3. Results

During the study period (2014–2021), a total of 11,814,508 DDD of oral antimicrobials were dispensed, prescribed by 349 primary care consultants (312 general practitioners and 37 pediatricians). The semestral post-intervention median study population was 342,086 inhabitants, compared to 335,046 inhabitants in the pre-intervention period (2014–2016).

Between 2017 and 2021, 6856 interventions were carried out, performing 3861 (56.3%) educational suggestions to optimize the use of antibiotics for UTIs. There was an average annual growth trend in interventions of 36.6%, interrupted only in 2020 as a result of the SARS-CoV-2 pandemic. Antibiotic modification or suspension in UTIs under recommendation was made in 2311 cases (74.3%).

### 3.1. Impact of Antibiotic Consumption

Throughout the study period, penicillins were the most prescribed antibiotics (66.0%). The NRA (co-amoxiclav, cephalosporins, and quinolones) included in the study accounted for 45.1% of the total. The temporal evolution in DID of the global prescription of any antimicrobial, including the NRA, in any period, is shown in Figure 2. The consumption of antibacterials in community settings in DID decreased by 33.7% between 2017 and 2021. The average DID between periods with the SD declined by −0.095 (0.325) (*p* < 0.0001). Likewise, the NRA group also showed a significant decrease of 39.1%, from 1126 (0.093) in 2014–16 to 0.686 (0.016) in 2017–21 (mean difference −0.439, [95% CI −0.342 to −0.536], *p* < 0.0001). The semestral changes in the consumption of beta-lactams, combined with beta-lactamase inhibitors, quinolones, and cephalosporins, expressed as DID, in the pre- and post-intervention period in the Lleida Health Care Region, are identified in Figure 3.

Table 1 illustrates the changes in antimicrobial consumption in three different points of the intervention (initial, middle, and last semesters) and the final impact. There was a substantial fall in antimicrobials used in Gram-negative urinary tract infections, i.e., quinolones, cephalosporins, and co-amoxiclav. Prior to the intervention, a consistent downward trend in the dispensing of quinolones and co-amoxiclav had been observed. However, this decline was maintained in the post-intervention period, with an immediate drastic reduction after the implementation of the ASP. There was a significant mean fall of −0.019 (0.097) per semester, particularly in the NRA, with values of −0.022 (0.049) for co-amoxiclav, −0.021 (0.032) for quinolones, −0.003 (0.015) for cephalosporins, and −0.095 (0.325) (*p* < 0.001) for all antimicrobials, which remained practically unchanged until the end of the period.

Regarding the expected trends, the intervention was also associated with significant changes in prescription after the intervention, with additional significant decreases of −0.2 (95% CI −0.4 to −0.1), −0.4 (−0.5 to −0.3), and −0.4 (−0.5 to −0.3) (*p* < 0.001) in DID per semester for quinolones, cephalosporins, and co-amoxiclav, respectively. In contrast, recommended antibiotics (RA) (fosfomycin trometamol and nitrofurantoin) did not show an increase.

### 3.2. Impact on Antimicrobial Resistance

The antibiotic sensitivity was tested in 20,587 and 3991 clinical urinary samples of *E. coli* and *K. pneumoniae*, respectively, collected over 8.5 years. 

Table 2 shows the semi-annual comparison of microbiological resistance rates. The table shows a drop in resistance rates, which increase significantly for *E. coli* over time in the three sections of the period. Resistance to the studied antimicrobials was increased in the period prior to the intervention for both bacteria, but was only significant for co-amoxiclav. Thus, *E. coli* rates changed from 5.7% (76/1342) in 2014 to 20.5% (263/1289) in 2016, and *K. pneumoniae* rates changed from 6.8% (8/118) in 2014 to 13.2% (35/239) in 2016 (*p* < 0.001). After the start of the ASP and throughout the post-intervention period, resistance rates maintained a linear decreasing trend for all the antimicrobials in the two *Enterobacteriaceae* (*p* < 0.005). The proportion of *E. coli* resistant to ciprofloxacin, ESBL-producing, and co-amoxiclav decreased significantly between the two periods, by 32.6%, 17.3%, and 9.3% (*p* < 0.001), respectively. The same was true for *K. pneumoniae* resistant to co-amoxiclav and ciprofloxacin, decreasing by 24.7% (OR 0.70, [95 % CI 0.59 to 0.84], *p* < 0.001) and 47.8%, respectively, but only from the second half of the post-intervention period (OR 0.46, [95 % CI 0.28 to 0.77], *p* = 0.003).

Table 3 shows the changes in the ID of resistance at three points in the intervention and the overall impact. Within the type of resistance studied, only the ID per 1000 inhabitants/day of *E. coli* resistant to ciprofloxacin and co-amoxiclav and the ESBL-producing strains fell significantly during the intervention period compared to the previous one (Figure 4). These IDs decreased by −0.595 cases (95% CI, −0.596 to −0.593) in ciprofloxacin resistance, −0.102 (−0.103 to −0.101) in co-amoxiclav resistance, and −0.115 (−0.116 to −0.114) in ESBL-producing strains, with a relative reduction of 41.2%, 17.4%, and 31.0% (*p* < 0.0001) at the end of the 5-year period. The observed increase in ID of *E. coli* resistant to co-amoxiclav, per semester, in the pre-intervention period (0.824 cases; [95% CI, 0.761 to 0.932], *p* < 0.0001) showed a change in −0.126 cases (–0.86 to –0.151) after the implementation of the ASP, with a change in the slope of −0.009 cases per 1000 inhabitants/day per semester (*p* < 0.001).

In the case of *K. pneumoniae*, no decreases in ID between periods were observed. However, when the intervention period was analyzed separately, a decline at the end of the period was observed (*p* < 0.0001). The observed increase in ID of *K. pneumoniae* resistant to co-amoxiclav, per semester, in the pre-intervention period (0.030 cases; [95% CI, 0.016 to 0.105], *p* < 0.0001) showed a change in −0.025 cases (−0.013 to −0.043) on the third semester of intervention (*p* < 0.001), with a change in the slope of −0.002 cases per 1000 inhabitants/day per semester (*p* < 0.001).

In neither bacteria did we observe an immediate change in trend in the ID, except for *E. coli* resistant to ciprofloxacin, with −16.4% (relative risk; RR 0.85, 0.83 to 0.84), and the ESBL-producing strains, with –20.4% (0.80, 0.79 to 0.80) (*p* < 0.0001), which occurred in the seventh semester (95% CI 6 to 12) (*p* < 0.05), concurrently with the start of the intervention. However, *K. pneumoniae* delayed the inflection point for the ESBL-producing strains until the penultimate semester by −34% (RR 0.66, 0.64 to 0.66), and ciprofloxacin and co-amoxiclav resistance in the last semester, by −35% (0.66, 0.65 to 0.67) and −38%, respectively (0.62, 0.61 to 0.63) (*p* < 0.0001).

### 3.3. Economic Impact

The average annual cost in antibacterials between 2016 and 2021 decreased from EUR 904,373 to EUR 548,836 (average difference—EUR 336,570), representing an absolute year-on-year difference in this period of −39.3%. The cost in euros per inhabitant changed from EUR 1.33 in the previous period to EUR 0.91 after the intervention, with an attributable cost saving of EUR 0.43/inhabitant/day.

## 4. Discussion

To our knowledge, this is the only study in our setting that evaluates the impact of a community ASP on microbial resistance over a long-term period (5 years), after an intervention regarding global antibiotic consumption and which caused an improvement in the adequacy of prescription. According to the observed trends, we found reductions of around 34% in the global dispensing of antimicrobials between 2017 and 2021 in the 23 basic health areas, which served 340,000 patients. Reductions in the dispensing of antibiotics considered harmful to the environment (quinolones, cephalosporins, and co-amoxiclav) were associated with a positive reduction in the *Enterobacteriaceae E. coli* and *K. pneumoniae* resistant to them, maintaining these results over the 5 years of the study. Our findings are similar to those reported by Boel [31] and Livermore [32], although their work was targeted at the study of the decline in cephalosporin and fluoroquinolone use. Interestingly, Livermore reported increased rebound use of beta-lactams/beta-lactamase inhibitors, which did not occur in our study.

Our results support the concept that reduced exposure to antibiotics, particularly quinolones, has a positive long-term effect on resistance patterns. The use of quinolones has been extremely high for decades due to their excellent oral bioavailability, pharmacokinetics, and their spectrum of activity against Gram-negative pathogens. However, in recent years, they have been subject to increasing safety concerns and regulatory alerts due to serious undesirable side effects, as well as the induction and alarming increase in resistance rates [33,34].

On the other hand, the additional evidence in this study is that ASPs could have a positive impact on the resistance rate of pathogenic microorganisms over time, regardless of the antibiotics, if the overall consumption is reduced, as some studies have described [35]. In our case, the general prescription of antibiotics suffered a progressive decrease that was remarkable in the last 2 years of the study, probably due to the low consumption in the second quarter of 2020. This drastic decrease coincided with the beginning of the COVID-19 pandemic, associated with modifications in the healthcare system, but has been maintained until today, with a sustained downtrend of DID in the community, as is also reflected in some international and national publications [36].

The dispensing of NRA decreased significantly, while fosfomycin trometamol and nitrofurantoin did not increase accordingly. This is despite the fact that they are the alternatives recommended in the P-ILEHRDA program, and it is in contrast to other published research findings [37]. This supports the possibility of the high and inappropriate prescription of various antimicrobials in situations such as asymptomatic bacteriuria or aseptic pyuria, where they generally should be avoided [38,39]. Moreover, it is unclear whether the use of fosfomycin trometamol is associated with decreased incidence of ESBL-producing *Enterobacteriaceae* [40]. However, its exclusive use in humans and its only indication as a single-dose treatment for uncomplicated UTIs would hinder the emergence of resistant mutants or the spread of plasmid-mediated resistance [41]. More research is needed to determine the role of fosfomycin trometamol in this regard. Likewise, nitrofurans also appear to have a similar relatively benign impact [38].

While most studies have focused exclusively on respiratory tract infections [42,43], our ASP also included the most prevalent infections in the community setting, with the main one being UTIs. Thus, an educational program with a comprehensive approach has been used, based on continuous training and periodic feedback of results to the team members, providing a strong positive reinforcement to continue participating, as indicated in other publications [44] and the recommendations of experts and societies [15,26]. In order to continue with the benefit of the effects obtained during the possible disruption caused by COVID-19, our ASP remained operational by preserving the training and broadcasting informational approaches through the new virtual communication systems that have emerged in the healthcare setting [45]. Restrictive interventions were not included as they are ineffective in modifying long-term behaviors and are poorly valued by professionals [46].

Our study linked antibiotic dispensing data with more than 25,000 positive urine samples for both *Enterobacteriaceae*, routinely collected, which provided, in this study, the adequate power to detect the direct relationship between consumption and resistance.

After 5 years from the beginning of the ASP, the IDs of *E. coli* resistant to ciprofloxacin and co-amoxiclav, and the ESBL-producing strains, were 41.2%, 17.4%, and 31.0% lower than the expected values in the pre-intervention period, respectively, as indicated by other studies with similar designs [37,47,48]. We were surprised that the associations were detectable within the first few months and were sustained significantly over time. However, this situation has already been described. An Israeli study evaluated the prevalence of ciprofloxacin resistance before, during, and after a nationwide restriction on quinolone use [49]. The study reported a reduction in resistance levels in the first month. Peñalva et al. [37] also demonstrated a decreasing trend after an intervention that began shortly after (3 to 6 months) and strengthened over time. Moreover, recent research has reported an association between higher rates of quinolone-resistant *E. coli* UTIs in a community with the increased use of antibiotics, regardless of prior individual antibiotic consumption [50]. According to the results of our investigation, this aspect may be relevant. We believe that the challenge of antibiotic resistance in the community setting is also inherent in the indirect effects of exposure, i.e., exposed people could transmit resistant bacteria to unexposed individuals, as Bell et al. have pointed out [4]. This could justify the similar reduction observed for the NRAs in the ID of *K. pneumoniae* within the period of action of the ASP, although it was not significant compared with the previous period. This aspect does not appear to have been previously described for these bacteria. All the above-mentioned evidence validates the crucial role of the actions adopted by our ASP in avoiding antimicrobials and maintaining the beneficial effects on our community setting.

The simple economic analysis based on commercial reference prices points to a cost reduction close to half of the amount previously reported. This amount, together with that obtained by our regional healthcare system, through a single regional cross-sectional ASP composed of acute care hospitals, long-term care facilities, geriatric residences, penitentiary, community pharmacists, dentists, podiatrists, and veterinarians, and the monetary data offered by the hospital setting [24], favor the sustainability of the health system and support the investment in more resources for the ASP [51]. Although it was not the subject of this study, the presumed avoidance of undesirable effects associated with medications and the hospitalizations derived from this fact could further increase savings.

This program is reproducible and generalizable to other community healthcare systems, for its ease of use and simplicity, as it does not require large human resources and assumes an acceptable workload. The education acquired by the professionals after the consultancies, the continuous training, and the easy access to the protocols, as tools for better prescriptions, are the main aspects associated with the great performance of this ASP.

Finally, the results of our study reveal other strengths. Firstly, we measured the use of dispensed rather than prescribed antibiotics. We consider this to be a strong measure of exposure and consumption, as it faithfully reflects the treated patients. Secondly, the work has not been affected by intercurrent interventions that could have created confusion in the interrupted temporal analysis of series. Prior to the start of the procedures of this ASP, the documentary aspects (guides/protocols) were created and distributed and formal training for the consultants was carried out. Thirdly, relying on an exclusive central microbiology laboratory avoided biased measurements in the urinary samples studied. On the other hand, our work has some limitations: first of all, only community data on general consumption and selected antibiotics were collected, while antimicrobials were also prescribed in acute care hospitals that already were undergoing a similar ASP, which could magnify the results in reducing resistance. Second, co-amoxiclav and quinolone prescription rates were already declining prior to the ASP implementation and thus we may have overestimated its impact on the already declining trend. Thirdly, our study focused on urinary *E. coli* and *Klebsiella* as the most common/predominant urinary pathogens. However, there are a wide variety of other organisms that were not counted. Finally, there was an absence of evaluation indicators to analyze the clinical and health impact, although process indicators on the quality of prescription to measure the effect of the program were analyzed.

## 5. Conclusions

The results of this study showed that, after 5 years, the implementation of an educational advisor community ASP was associated with outstanding benefits in the reduction of antimicrobial consumption and urinary *E. coli* and *K. pneumoniae* resistance, with an associated smaller economic cost.

## Figures and Tables

**Figure 1 antibiotics-11-01776-f001:**
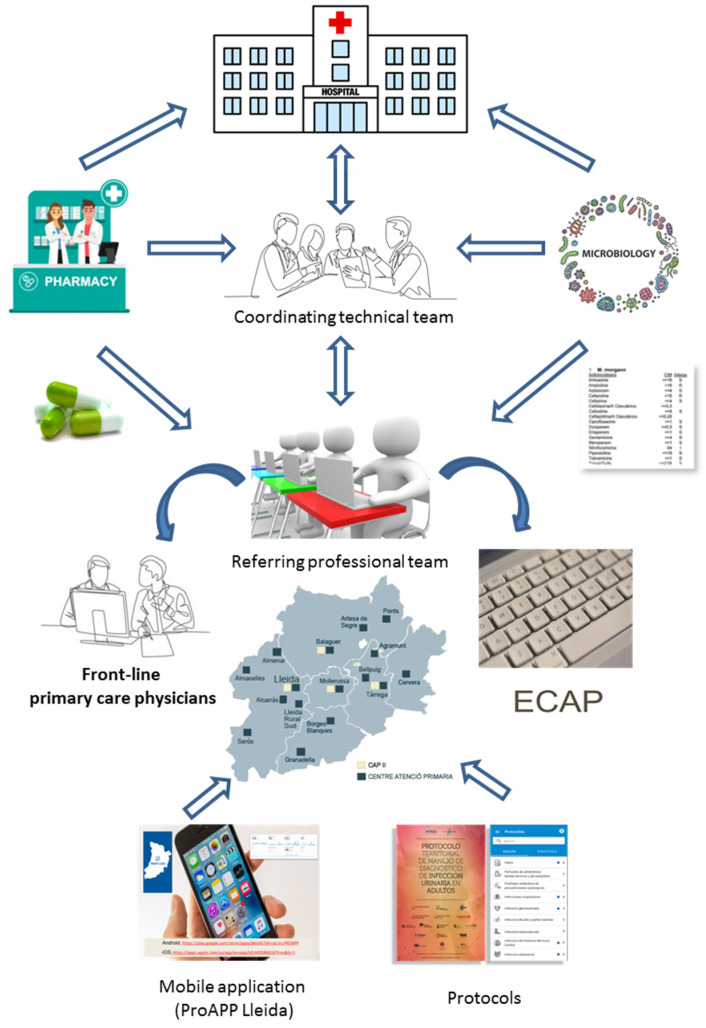
Organizational flow chart.

**Figure 2 antibiotics-11-01776-f002:**
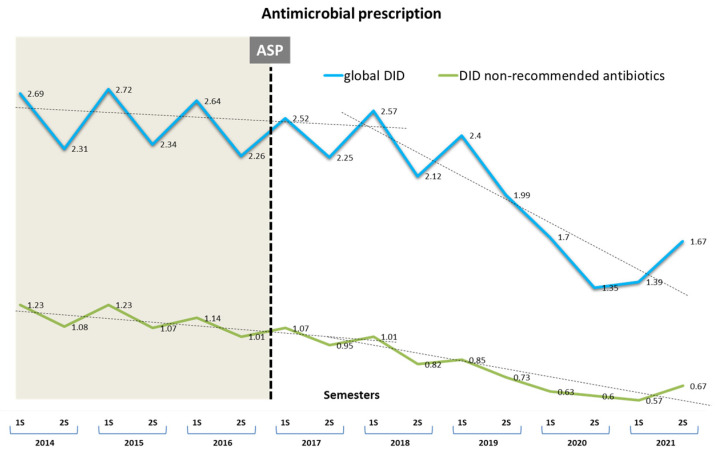
Semestral evolution of defined daily doses per 1000 inhabitants/day (DID). Global antimicrobial prescription (upper line) and non-recommended antimicrobials (NRA) (lower line). Shaded area represents the pre-antibiotic stewardship program (ASP) intervention period.

**Figure 3 antibiotics-11-01776-f003:**
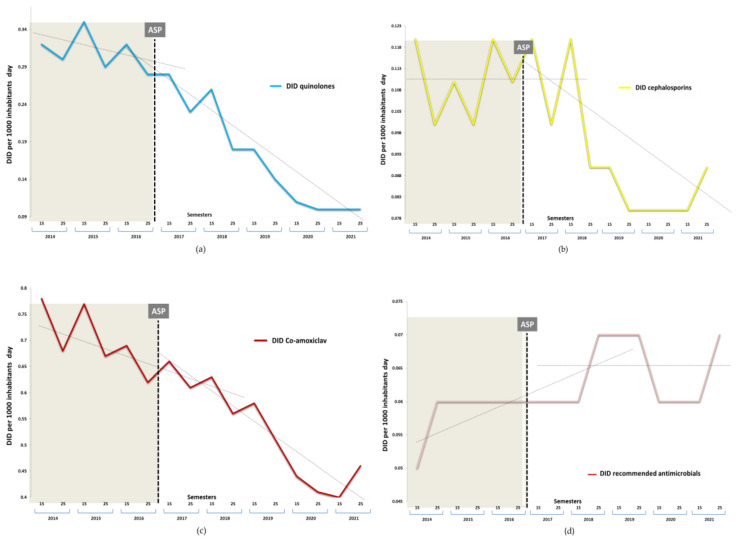
Semestral evolution of defined daily doses per 1000 inhabitants/day (DID). (**a**) Quinolones, (**b**) cephalosporins, (**c**) co-amoxiclav, (**d**) recommended antimicrobials: fosfomycin trometamol, nitrofurantoin.

**Figure 4 antibiotics-11-01776-f004:**
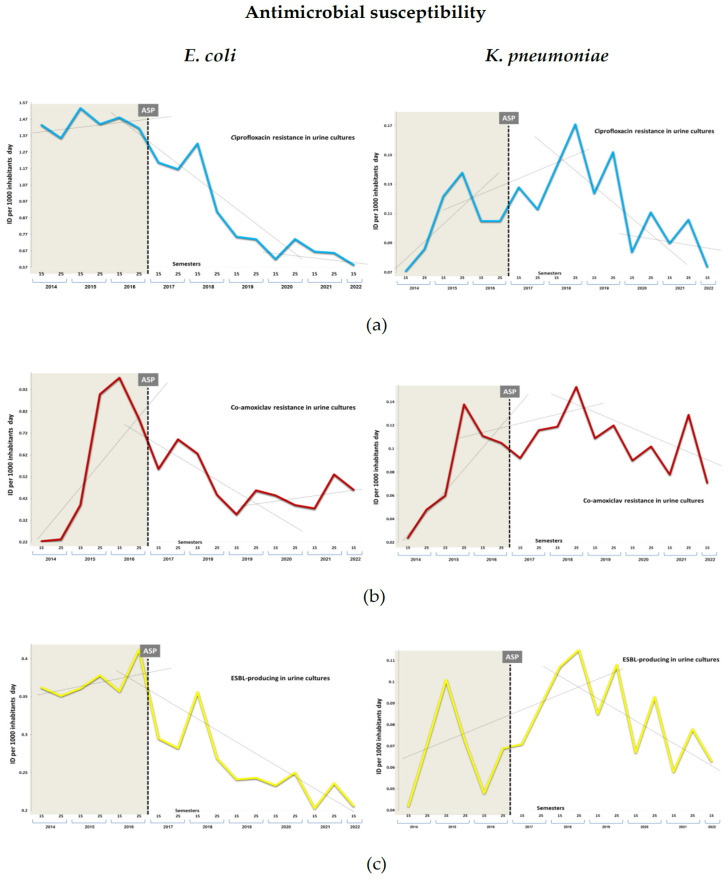
Semestral evolution of antibiotic-resistant *E. coli* and *K. pneumoniae* in urine cultures per incidence density (ID) per 1000 inhabitants/day. (**a**) Ciprofloxacin resistance, (**b**) co-amoxiclav resistance, (**c**) extended-spectrum B-lactamase (ESBL).

**Table 1 antibiotics-11-01776-t001:** Changes in antimicrobial prescription and final impact.

Prescribed Antibiotic	DID Pre-Intervention Period	Relative Change First Semester 2017	Relative Change First Semester 2019	Relative Change Second Semester 2021	Absolute Effect Post-Intervention Period	Relative Effect (%)
**Total antibiotics** **(J01)**	2.496	0.892 (0.890 to 0.894)	0.790 (0.787 to 0.781)	0.670 (0.668 to 0.672)	−0.688 (−0.691 to −0.685)	27.57 (27.65 to 27.49)
**Total of non-recommended antibiotics (NRA)**	1.126	0.842 (0.839 to 0.845)	0.649 (0.646 to 0.652)	0.583 (0.581 to 0.586)	−0.412 (−0.414 to −0.410)	36.59 (36.70 to 36.48)
**Co-amoxiclav** (J01CR02)	0.704	0.866 (0.862 to 0.870)	0.726 (0.722 to 0.730)	0.660 (0.657 to 0.663)	−0.227 (−0.229 to 0.226)	32.27 (32.42 to 32.12)
**Quinolones** (J01M)	0.311	0.754 (0.748 to 0.759)	0.439 (0.435 to 0.443)	0.328 (0.325 to 0.312)	−0.158 (−0.159 to −0.157)	50.91 (51.09 to 50.74)
**Ciprofloxacin** (J01MA02)	0.114	0.779 (0.770 to 0.788)	0.556 (0.549 to 0.564)	0.439 (0.433 to 0.446)	−0.052 (−0.052 to −0.051)	45.39 (45.70 to 45.07)
**Levofloxacin** (J01MA12)	0.132	0.828 (0.819 to 0.837)	0.457 (0.451 to 0.464)	0.338 (0.332 to 0.343)	−0.061 (−0.061 to 0.060)	45.89 (46.18 to 45.60)
**Cephalosporins** (J01D)	0.111	0.936 (0.925 to 0.947)	0.749 (0.740 to 0.758)	0.807 (0.798 to 0.817)	−0.026 (−0.027 to −0.026)	23.90 (24.31 to 23.49)
**Cefuroxime** (J01DC02)	0.061	0.739 (0.726 to 0.751)	0.614 (0.603 to 0.625)	0.433 (0.424 to 0.442)	−0.025 (−0.025 to −0.025)	40.96 (41.41 to 40.50)
**Third-generation cephalosporins** (J01DD)	0.046	1.241 (1.222 to 1.261)	0.935 (0.919 to 0.951)	1.275 (1.256 to 1.295)	0.0003 (−0.0003 to 0.0004)	0.07 (−0.73 to 0.87)
**Total** **recommended antibiotics (RA)**	0.059	1.107 (1.092 to 1.123)	1.267 (1.250 to 1.284)	1.130 (1.114 to 1.146)	0.0001 (−0.0004 to 0.0005)	0.10 (−0.60 to 0.80)
**Fosfomycin trometamol** (J01XX01)	0.042	1.026 (1.007 to 1.044)	1.137 (1.118 to 1.156)	1.150 (1.131 to 1.169)	−0.001 (−0.002 to −0.001)	3.15 (3.96 to 2.33)
**Nitrofurantoin** (J01XE01)	0.018	1.300 (1.268 to 1.332)	1.574 (1.539 to 1.610)	1.082 (1.054 to 1.111)	0.001 (0.001 to 0.002)	7.21 (6.03 to 8.39)

DID; defined daily dose per 1000 inhabitants per day with 95% CIs, unless otherwise specified. NRA; non-recommended antibiotics (co-amoxiclav, quinolones, cephalosporins). RA; recommended antibiotics (fosfomycin trometamol, nitrofurantoin).

**Table 2 antibiotics-11-01776-t002:** Rates of microbiological resistance.

	Antimicrobial Resistance	Comparisons by Semester (S)													
		Second S 2016 vs. Second S 2017				Second S 2016 vs. Second S 2019					Second S 2016 vs. First S 2022				
		% (n/N) Pre- Intervention Resistance	% (n/N) Post- Intervention Resistance	OR CI 95%	*p*	% (n/N) Pre- Intervention Resistance	% (n/N) Post- Intervention Resistance	OR CI 95%	*p*	Prevented Rate (%)	% (n/N) Pre- Intervention Resistance	% (n/N) Post- Intervention Resistance	OR CI 95%	*p*	Prevented Rate (%)
** *E. coli* **	ESBL-producing	10.73 (138/1285)	7.69 (96/1248)	0.68 (0.52–0.90)	0.007	10.73 (138/1285)	7.11 (83/1167)	0.63 (0.47–0.84)	<0.001	33.8 (14.1–48.5)	10.73 (138/1285)	6.13 (72/1174)	0.54 (0.40–0.73)	<0.001	45.7 (27.8–59.1)
	Ciprofloxacin	36.89 (474/1285)	31.81 (397/1248)	0.79 (0.67–0.94)	0.004	36.89 (474/1285)	21.67 (253/1167)	0.47 (0.39–0.56)	<0.001	41.2 (33.0–48.4)	36.89 (474/1285)	17.46 (205/1174)	0.36 (0.29–0.43)	<0.001	52.7 (45.3–59.0)
	Co-amoxiclav	20.46 (263/1285)	18.72 (233/1248)	0.89 (0.73–1.09)	NS	20.46 (263/1285)	13.36 (156/1167)	0.59 (0.48–0.74)	<0.001	34.7 (21.7–45.5)	20.46 (263/1285)	13.71 (161/1174)	0.61 (0.49–0.76)	<0.001	33.0 (19.8–44.0)
** *K. pneumoniae* **	ESBL-producing	9.62 (23/239)	13.95 (30/215)	1.52 (0.88–2.71)	NS	9.62 (23/239)	14.17 (37/1167)	1.05 (0.89–2.69)	NS	NA	9.62 (23/239)	8.80 (22/250)	0.90 (0.49–1.67)	NS	NA
	Ciprofloxacin	14.64 (35/239)	17.67 (38/215)	1.25 (0.75–2.06)	NS	14.64 (35/239)	19.92 (52/1167)	1.45 (0.90–2.32)	NS	NA	14.64 (35/239)	10.40 (26/250)	0.67 (0.39–1.16)	NS	NA
	Co-amoxiclav	14.64 (35/239)	18.13 (39/215)	1.29 (0.78–2.12)	NS	14.64 (35/239)	15.70 (41/1167)	1.08 (0.66–1.77)	NS	NA	14.64 (35/239)	10.00 (25/250)	0.64 (0.37–1.11)	NS	NA

(n/N); n; total positive antibiograms, N; total antibiograms, ESBL; extended-spectrum β-lactamase. Co-amoxiclav; amoxicillin–clavulanic acid; NS; not significant. NA; not applicable.

**Table 3 antibiotics-11-01776-t003:** Changes in incidence density and final impact.

Antimicrobial Resistance	ID Pre-Intervention Period	Relative Change Second Semester 2017	Relative Change Second Semester 2019	Relative Change First Semester 2022	Absolute Effect Post-Intervention Period	Relative Preventable Effect (%)
** *E. Coli* **						
**ESBL-producing**	0.370 (0.369 to 0.371)	0.762 (0.757 to 0.768)	0.656 (0.652 to 0.662)	0.556 (0.552 to 0.560)	−0.115 (−0.116 to −0.114)	31.01 (31.22 to 30.80)
**Ciprofloxacin**	1.445 (1.443 to 1.447)	0.808 (0.805 to 0.810)	0.513 (0.511 to 0.515)	0.406 (0.404 to 0.407)	−0.595 (−0.596 to −0.593)	41.20 (41.29 to 41.10)
**Co-amoxiclav**	0.583 (0.582 to 0.585)	1.186 (1.181 to 1.191)	0.783 (0.779 to 0.787)	0.789 (0.785 to 0.793)	−0.102 (−0.103 to −0.101)	17.46 (17.65 to 17.26)
** *K. pneumoniae* **						
**ESBL-producing**	0.067 (0.067 to 0.067)	1.326 (1.310 to 1.343)	1.614 (1.595 to 1.632)	0.937 (0.923 to 0.950)	0.018 (0.017 to 0.018)	20.84 (20.34 to 21.35)
**Ciprofloxacin**	0.104 (0.104 to 0.105)	1.080 (1.069 to 1.092)	1.458 (1.445 to 1.472)	0.712 (0.702 to 0.721)	0.013 (0.012 to 0.014)	11.13 (10.67 to 11.59)
**Co-amoxiclav**	0.081 (0.080 to 0.081)	1.437 (1.422 to 1.453)	1.490 (1.474 to 1.506)	0.887 (0.875 to 0.899)	0.027 (0.026 to 0.027)	24.81 (24.38 to 25.25)

ID; incidence density per 1000 inhabitants per day expressed with 95% CIs, unless otherwise specified. ESBL; extended-spectrum β-lactamase. Co-amoxiclav; amoxicillin–clavulanic acid.

## Data Availability

Not applicable.

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
