# Peer review of "Impact of a Primary Care Antimicrobial Stewardship Program on Bacterial Resistance Control and Ecological Imprint in Urinary Tract Infections"

_antibiotics, 2022, doi:10.3390/antibiotics11121776_

Round 1

Reviewer 1 Report

This article describes the impact and feasibility of an antimicrobial stewardship program at the outpatient/ambulatory level. This is not a novel concept per say; but it implies proof of feasibility in an real world setting.

I recommend utilizing English language editing services since multiple sentences were really difficult to follow.

Abstract: Lines 62-63: there is an abundance of literature that describes reduction in resistance after the implementation of antibiotic stewardship programs. A brief search on PubMed brings up several hundred results of ASP successes; as a few examples:
- Effect of antibiotic stewardship interventions in primary care on antimicrobial resistance of Escherichia coli bacteraemia in England (2013–18): a quasi-experimental, ecological, data linkage study; DOI:https://doi.org/10.1016/S1473-3099(21)00069-4
- Impact of an Antimicrobial Stewardship Program on Antimicrobial Utilization, Bacterial Susceptibilities, and Financial Expenditures at an Academic Medical Center; PMID: 27803499
- Impact of antimicrobial stewardship programs on antibiotic consumption and antimicrobial resistance in four Colombian healthcare institutions; DOI: https://doi.org/10.1186/s12879-022-07410-6
- Relationship of carbapenem restriction in 22 university teaching hospitals to carbapenem use and carbapenem-resistant Pseudomonas aeruginosa; PMID:19273670
- Implementing an infection control and prevention program decreases the incidence of healthcare-associated infections and antibiotic resistance in a Russian neuro-ICU; PMID: 30083313

The abstract is unclear; it needs to be more succinct --- in a few statements summarize the intervention and the major outcome findings. You do not need confidence intervals in the abstract. Your abstract distills to 1. AMSP have been a central component in reducing over prescription of unnecessary antibiotics with multiple studies showing a corresponding benefit in reduction of bacterial resistance. 2. Little data/less commonly AMSP have been studied in outpatient ambulatory settings. 3. Your group implemented an AMSP in a large regional outpatient setting to assess feasibility and effectiveness. 4. Over a 5-year post-implementation period, compared to the pre-intervention period, a significant reduction in antibiotic prescription occurred with a corresponding reduction in resistance in E. coli urinary isolates. 5. AMSP activities also was found to be cost effective with a reduction in attributable medication expenditure.

Introduction:
The first lines are really sensationalized – “superbugs” are not an actual medical term, and the reference highlights that “analysis undertaken by this Review showed that of the 10 million total deaths that might be associated with drug resistance each year by 2050, around a quarter will come from drug-resistant strains of TB” --- so you are including tuberculosis (which is not what you’re investigating) with the global estimate encompassing underserved/resource limited settings with their own confounding challenges. The current data on increasing antibiotic resistance epidemiology is compelling enough to describe the scope of the problem.

Lines 89 – 90: Antimicrobial stewardship plays a role in reducing “emergencies” (?) and “transmitting resistant pathogens” (?) ---- antibiotic stewardship has been associated with reducing local antibiogram and community resistance rates but not directly with patient-to-patient transmission. You may be combining antibiotic resistance and infection control measures (which are often bundled programs)

Lines 98 – 105: Use specifics: what is the current rate of ESBL-Enterobacteriaceae in epidemiologic studies and how have they changed over the previous years. Regarding increased mortality – what are some studies that highlight the differences in mortality with MDR/ESBL v. wild-type organisms
“Alarming” prevalence/increases in Spain of ESBL-phenotype --- what are these numbers/are the rates of ESBL GNR/E. coli increasing faster in Spain compared to the rest of Europe

I do not think you need to discuss E. coli and Klebsiella as the main cause of urinary tract infection ----> this is more for part of your methods about why you chose specifically to focus on these organisms as part of your study methodology

Lines 106-113: I am not sure what you are trying to say; cephalosporins/beta-lactams facilitate selection for ESBL organisms? IDSA guidelines recommend restricting fluroquinolones (?) to protect ecosystem --- I think the priority is always to use the narrowest spectrum agent.

Lines 120 – 125: (?) “anomalous context”

Lines 126 – 129 are unclear: Briefly restate the major findings of the prior study that you published and are referencing. Perhaps “We sought to expand the scope of a successful AMSP to the regional outpatient ambulatory setting with a specific focus on reducing the duration of antibiotics for urinary tract infection and curbing prescription of fluroquinolones, cephalosporins, and amoxicillin-clavulanic acid. Our hypothesis was that this program would be feasible in a large healthcare network and result in meaningful reduction in antimicrobial resistance in E. coli and Klebsiella and reduced healthcare costs.”

Section 2 should be methods ---- It was impossible to understand the results/without understanding the elements of the AMSP implemented. The entire methods section needs to be very clear as to what the intervention and AMSP entailed.

Regarding the Methods Section Page 12-14
Setting: describe more clearly the regional healthcare setting for readers that are not familiar with it
-- You highlight the regional population; are all primary care centers integrated? Does this encompass all outpatient/primary care for the region or are there other independent practitioners/urgent care/clinics that are not assessable/censored?

-- Intervention – this is part of design

-- Section heading: AMSP design (also FYI AMSP program design = Antimicrobial stewardship program program <Repeated word> design)] A flow sheet/graphical representation of the programs elements/workflow would be really helpful here
This section needs to be clear and specific because it is the cornerstone of your manuscript
- how often was training of GP/primary care performed/was there a set schedule
- Are you indicating that EVERY positive microbiologic culture results was reviewed by a physician (or who reviewed this) – and then daily-advisory recommendations were sent to front-line primary care physicians? That is a huge workflow/huge volume for review. Were these recommendations extended to ALL antibiotic prescriptions --- or just urinary tract infections? Does this imply that culture-negative specimens were not reviewed/and thus a source of potential bias for empiric regimens/not addressable by the stewardship initiative?

--- Evaluation methods --- to clarify, for incident density --- would a single person be able to be counted multiple times per semester (i.e. if a person had two or three positive urine cultures/UTI in a 6-month period would they be counted more than once?)

RESULT section:
- Lines 130 – 141: based on what is written – the entirety of the region had a total of 349 primary care physicians; does this capture the entirety of the population?
The DDD of 11,814,508 --- I assume this was the pharmacy dispensing record --- so this would include ALL oral antibiotics for all causes (including respiratory/post-hospitalization/urinary tract infection etc?)
It is not clear, but the way I am interpreting it is that 6,856 AMSP daily advisory recommendations were made with 56% of these recommendations for urinary tract infection?

Lines 167 – 171: “global impact?”
Lines 170 – 171 seems to suggest that amox-clav and quinolone prescription rate was already declining PRIOR to the AMSP’s implementation; if this is true then is an important point and also is a limitation in your data in that AMSP may be over-estimated in its impact to already prevailing trend
Table 1: using discrete selected semesters makes appear that favorable time periods were chosen --- the table would be better if you were comparing annual differences: the year 2016 (prior to implementation of AMSP) compared to 2017, 2018, 2019, 2020, 2021

Table 2: Again, why not compare resistance in 2016 sequentially versus each year as opposed to selected “2nd semester 2017/2019/2022”; also you need an N number, how many isolates and what percentage resistant
i.e. as an extreme example --- 6% resistance in 1000 isolates v. 10% in 100 isolates

Lines 206 – 233: This is hard to understand. It would be better to describe the resistance changes in E. coli through the study period separately and THEN the results in Klebsiella as opposed to combining them together

General points on the discussion session:
There needs to be a better discussion on limitations: your study focused on urinary E. coli and Klebsiella as the most common/predominant urinary pathogens ---- however, there is a wide variety of other organisms that are not counted
Daily doses of antibiotics --- you have inherently skewed your results because a single uncomplicated UTI course with Fosfomycin is 1 dose (1 DDD) as opposed to ciprofloxacin which would be 3 days (3 DDD). Accordingly, the entire set up for preferred antibiotics are skewed towards showing a reduced DDD when total antibiotic treatment courses may be more representative

The discussion section implies that while antibiotic use of cephalosporins/amox-clav/quinolone decreased there was NOT a corresponding increase in either Macrobid or Fosfomycin ---- this indicates that LESS overall cases of urinary cultures were being treated. Is this true --- because the AMSP recommendations seem to favor utilizing Fosfomycin/nitrofurantoin > other antibiotics so this should have resulted in some increase in prescription of these preferred agents when compared to the pre-intervention period. Are you suggesting that that the major impact of the AMSP was reduction of treatment for what was asymptomatic bacteruria

The biggest points that should be part of the discussion section is whether this program is applicable/extrapolatable to other healthcare systems --- especially as the staff/resources to do individual daily audit of all cultures may not be feasible and the infrastructure may not be available to provide real-time audit recommendations to all primary care physicians; instead, what are the highest yield aspects of an AMSP. The discussion should not be a restatement of the results, but an interpretation of their meaning and the key take away points.

Reviewer 2 Report

This is a well executed study over a period of 5 years on huge number of clinical samples. Please find my comments below:-

1. Conclusion should be modified with key message.

2. Many typo and grammatical mistakes were identified.

3. English language of the text should be improved drastically.

Reviewer 3 Report

Please, find the review attached.

Round 2

Reviewer 3 Report

Thanks for the detailed comments